# Ice sheet and precession controlled subarctic Pacific productivity and upwelling over the last 550,000 years

Zhengquan Yao [1,2,9] ✉, Xuefa Shi [1,2,9] ✉, Qiuzhen Yin [3] ✉, Samuel Jaccard [4], Yanguang Liu [1,2], Zhengtang Guo [5], Sergey A. Gorbarenko [6], Kunshan Wang[1,2], Tianyu Chen [7], Zhipeng Wu[3], Qingyun Nan[8], Jianjun Zou[1,2], Hongmin Wang[1,2], Jingjing Cui[1,2], Anqi Wang[1,7], Gongxu Yang[1], Aimei Zhu[1,2], Aleksandr Bosin [6], Yuriy Vasilenko [6] & Yonggui Yu[1,2]

The polar oceans play a vital role in regulating atmospheric $CO_2$ concentrations ($p$CO$_2$) during the Pleistocene glacial cycles. However, despite being the largest modern reservoir of respired carbon, the impact of the subarctic Pacific remains poorly understood due to limited records. Here, we present high-resolution, $^{230}$Th-normalized export productivity records from the subarctic northwestern Pacific covering the last five glacial cycles. Our records display pronounced, glacial-interglacial cyclicity superimposed with precessional-driven variability, with warm interglacial climate and high boreal summer insolation providing favorable conditions to sustain upwelling of nutrient-rich subsurface waters and hence increased export productivity. Our transient model simulations consistently show that ice sheets and to a lesser degree, precession are the main drivers that control the strength and latitudinal position of the westerlies. Enhanced upwelling of nutrient/carbon-rich water caused by the intensification and poleward migration of the northern westerlies during warmer climate intervals would have led to the release of previously sequestered $CO_2$ from the subarctic Pacific to the atmosphere. Our results also highlight the significant role of the subarctic Pacific in modulating $p$CO$_2$ changes during the Pleistocene climate cycles, especially on precession timescale (~20 kyr).

Atmospheric carbon dioxide concentrations ($p$CO$_2$) recorded in ice cores have varied cyclically with oscillations typically ranging from ~180 ppmv during glacials to ~280 ppmv during interglacials[1]. Yet, the mechanisms responsible for the variations in $p$CO$_2$ remain elusive. Previous studies have proposed the Southern Ocean as a key player in modulating $p$CO$_2$ fluctuations over glacial-interglacial cycles[2-4]. However, recent literature highlights a notable gap in our understanding, particularly concerning the subarctic Pacific which is regarded as the

polar twin of the Southern Ocean[5]. The North Pacific is the largest reservoir of respired carbon, comprising nearly half of the global ocean inventory today[6]. This substantial size of reservoir raises the possibility that it may have substantially influenced the carbon cycle in the past as well[7]. Recent researches highlight the substantial impact of the subarctic Pacific in regulating the rapid deglacial rise in $p$CO$_2$ levels[8-11], and imply that it may have been a crucial role over longer timescales in the Pleistocene[12-14]. Furthermore, recent research

challenges conventional wisdom by proposing that, during the latter stages of the last deglacial period, the North Pacific, rather than the Southern Ocean, may have contributed the lion's share of $CO_2$ degassing during the second half of the last deglacial[15].

The exchange of $CO_2$ between the atmosphere and the deep ocean carbon reservoirs, driven by a combination of physical and biological processes[2,8,10,16], is the primary modulator of glacial-interglacial $pCO_2$ changes. In this context, upwelling and vertical mixing in the North Pacific have been viewed as a crucial mechanism that influence Pleistocene $pCO_2$ fluctuations by regulating the release of deeply-sequestered $CO_2$ to the atmosphere[8–10,14]. To date, however, the mechanisms driving upwelling and, more broadly, the role of the subarctic Pacific in regulating Pleistocene $pCO_2$ variations remain poorly understood, due to the scarcity of high-resolution records spanning several glacial cycles. Indeed, only a few studies have addressed the mechanisms controlling the rate of upwelling in the subarctic Pacific, with a predominant focus on the last deglacial transition[10,17,18].

Export production proxies have been widely used to reconstruct changes in the rate of upwelling and vertical mixing[8,14]. Based on the productivity proxy, an upwelling record covering the past 850 kyr was reconstructed in the Bering Sea[14], yet a [230]Th normalization method is necessary to evaluate the efficacy of the productivity proxy[19]. Moreover, with multiple factors influencing upwelling in the subarctic Pacific[10,14,18], it is crucial to differentiate these elements and assess their respective contributions. This distinction is pivotal to reveal the mechanisms that govern upwelling. In this study, we present high-resolution, [230]Th-normalized export production and upwelling records from the subarctic Pacific, spanning the last five glacial-interglacial cycles. Based on the reconstructed productivity and upwelling records, in conjunction with transient climate model simulations, we explore the mechanisms related in particular to orbital forcing, greenhouse gasses and ice-sheet extent, underpinning the variations in productivity and upwelling on orbital timescales.

## Results and discussion
### Productivity changes over the past 550 kyr
Sediment core LV76-16-1 (48.85°N, 168.46°E, 2,374 m water depth) was recovered from the Emperor Seamount chain in the subarctic Pacific (Fig. 1). The 7.6 m long core is texturally homogeneous, dominated by fine silt and clay. The age model of core LV76-16-1 was established by assigning the abrupt increases in Ca/Ti to glacial terminations[20] (Supplementary Fig. 1a, b). This correlation is based on the observation that

each deglacial transition is characterized by transient increases in the sedimentary accumulation of biogenic carbonate all across the North Pacific[12,13,21]. Indeed, the downcore variability in Ca/Ti ratio of core LV76-16-1 is comparable to variations in the Ca/Al ratio and Ca counts at nearby ODP Site 882[13] and core MD2416[22] (Fig. 1; Supplementary Fig. 2), both of which have a well-constrained chronology. Specifically, the chronology for ODP Site 882 was established by correlating high-resolution X-ray fluorescence (XRF) scanning Ba/Al ratios with the millennial-suborbital variability of δD from Antarctica ice core[12,22]. The age model for MD2416 were developed by aligning XRF scanning Ca counts with those from ODP Site 882[22]. This demonstrates that the chronology of core LV76-16-1 has a high enough resolution to allow for discussions of suborbital-scale variations. The calculated sedimentary accumulation rates of core LV76-16-1 show a coherent downcore variability, typically ranging between ~1.2 and 1.7 cm/kyr (Supplementary Fig. 1c).

Paleoproductivity proxies such as biogenic barium (BioBa), biogenic silica (opal), and carbonate have been widely used to reconstruct past changes in biological productivity[23]. Calcium (Ca) normalized to Ti reflects the sedimentary concentration of biogenic carbonate ($CaCO_3$), and is consistent with the $CaCO_3$ content measured in discrete samples (Supplementary Fig. 1b). Similarly, Ba/Ti in marine sediments is often used to reflect the concentration of BioBa. Unlike opal and carbonate, BioBa is considered to reflect integrated export production from the photic zone[23], and can thus be considered as the sum of biogenic opal and carbonate, as silicious (diatoms) and calcareous (coccoliths) phytoplankton constitutes most of the primary producers in the subarctic Pacific[24]. This is corroborated by the good correspondence between the BioBa record and the normalized sum of the percentages of opal and $CaCO_3$ contents downcore (Supplementary Fig. 3a, b).

The sedimentary opal content and Ca/Ti and Ba/Ti ratios all display, coherent, cyclic variations throughout the past 550 kyr (Fig. 2). Comparison of their downcore variations with the benthic δ[18]O stack[20] shows that variations in these three proxies track glacial-interglacial cycles, with high values during interglacials and generally lower values during glacials (Fig. 2). This indicates that the temporal variations of these productivity proxies are strongly linked to glacial-interglacial cyclicity. In addition, comparison of the evolution of Earth's precession and obliquity parameters in the past with our export production records suggest that their variations are also controlled by orbital forcing. The Ca/Ti ratio corresponds well and positively with obliquity (Fig. 2a), while the opal content and the Ba/Ti ratio correlate negatively

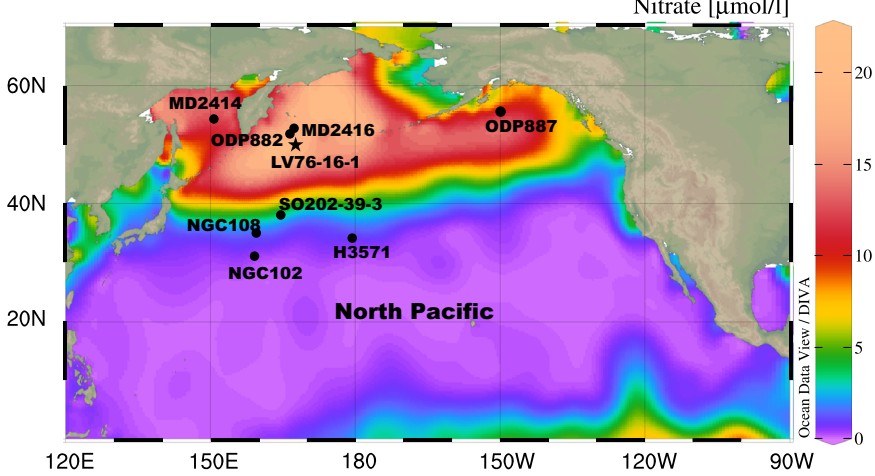

**Fig. 1 | Map showing general information about the subarctic North Pacific.** Modern nitrate concentration (µmol/l) of the surface water generated with Ocean Data View[65]. The star indicates the site of sediment core LV76-16-1. The locations of other sediment cores mentioned in the text (also the Supplementary Fig.) are indicated by solid circles. The base map is generated using Ocean Data View (http://odv.awi.de/).

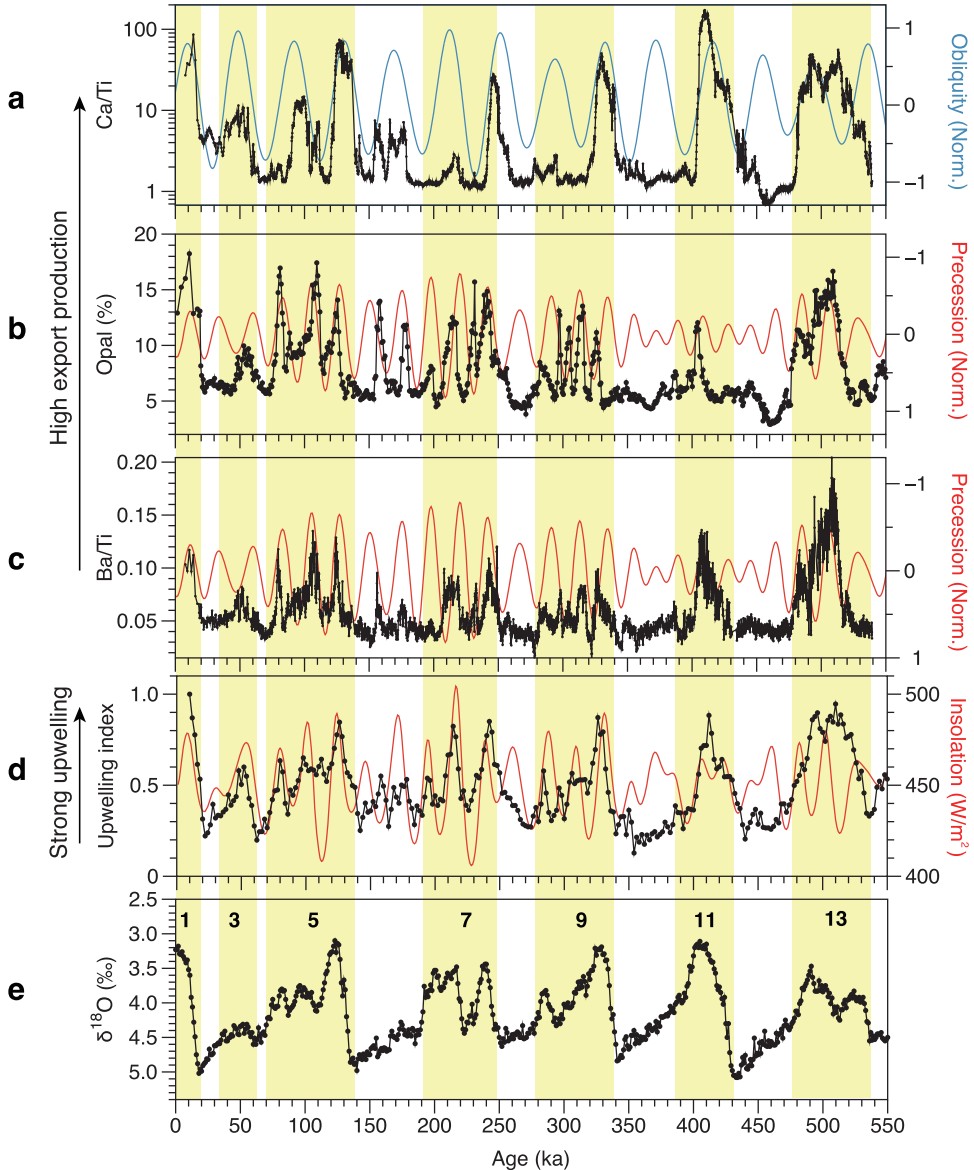

**Fig. 2 | Comparison of productivity and upwelling proxies of core LV76-16-1 with orbital parameters and the marine oxygen isotope record. a–c** Variations in Ca/Ti, opal and Ba/Ti plotted with normalized obliquity (blue line) and precession[60] (red line). **d** Reconstructed upwelling index of core LV76-16-1 and summer insolation at 65° in the northern Hemisphere[60] (red line). **e** Marine benthic $\delta^{18}O$ stack[20]. The vertical yellow bars represent interglacial periods with the corresponding marine isotope stages indicated. Source data are provided as a Source Data file.

with precession (Fig. 2b, c). It is noteworthy that the effect of orbital forcing was particularly pronounced during interglacials when the global ice volume was small, while it was generally suppressed during full glacial intervals (Fig. 2a–c). For example, the precession peaks during MIS 1, 5, 7, 9 and 13 are clearly visible in the opal and Ba/Ti records, yet they are more subdued during MIS 2, 6, 8, 10 and 12 (Fig. 2b, c). Further analysis shows that the precessional signal filtered out from the Ba/Ti record compares very well with precession throughout the last 550 kyr (Supplementary Fig. 4a), confirming the role of precession. Wavelet analysis shows that the variations of the Ba/Ti and upwelling records contain three major periodicities of ~100 kyr, ~40 kyr and ~20 kyr (Supplementary Fig. 4b, c). The power of the ~20 kyr cycle (so does the power of the ~40 kyr cycle) is varying in time mainly due to the amplitude modulation of precession by eccentricity. The weak precession and strong obliquity signals during MIS 11 in all productivity proxies may be related to the relatively low amplitude of precession variability, caused by low eccentricity and the large

variation in obliquity during MIS 11, a particularity which has also been observed in other climate variables[25,26].

Productivity proxies may be subject to differential preservation during diagenesis[21], due to the effects of degradation and dissolution[8,27], which may complicate their interpretation. However, the consistent variations between the Ba/Ti and the sum of normalized $CaCO_3$ and opal contents, along with clearly expressed orbital signals, indicate that dissolution of these biogenic components in the water column and the sediments did not substantially obscure the sedimentary records. Thus, the downcore BioBa record can be interpreted as a robust indicator of changes in integrated export production. This is further supported by the overall good agreement between the variations in sedimentary opal, $CaCO_3$ and BioBa concentrations with the $^{230}Th$-normalized fluxes of each proxy over the last glacial cycle (Supplementary Fig. 5), which can also be applied for the old portion of the record. The reconstructed productivity history of core LV76-16-1 is largely consistent with previous studies covering a large region of the subarctic Pacific, including the Bering Sea[28], Okhotsk Sea[29], NW[12,30] and

NE Pacific[31] (Supplementary Fig. 6a–d), all of which show higher export production during warm periods.

Export productivity in the subarctic Pacific can be enhanced by the alleviation of light limitation[32], enhanced supply of iron[33] and/or upwelling[10]. Previous results at nearby ODP Site 882 preclude the role of sea-ice-driven light limitation to account for generally reduced export production during glacial times, because the site, analogous to the location of core LV76-16-1, is located southeast of the summer sea-ice limit even during glacial maxima[12]. Enhanced productivity can also be caused by an increased iron supply associated with ice-rafted detritus (IRD) input via the melting of sea ice/icebergs[34], and/or aeolian transport of dust[35]. However, these factors can also be excluded, as the IRD variations in core LV76-16-1 are inversely correlated with changes in opal and BioBa concentrations (Supplementary Fig. 3c). On the other hand, previous study suggested that Fe input associated with increased dust supply during the glacials did not exert a substantial influence on export production[21]. Although the subarctic Pacific is a high-nutrient low-chlorophyll (HNLC) region (Fig. 1), biological productivity in pelagic ecosystems is primarily sustained by the supply of nutrients, including Fe, to the euphotic zone via upwelling. Therefore, biological productivity in this region may have been primarily driven by changes in the upwelling of nutrient-rich subsurface water. This inference is consistent with previous studies in the subarctic Pacific covering various timescale[9,10,12,14]. Our results suggest that warm interglacial conditions and high boreal summer insolation, resulting from high obliquity and low precession, conspire to sustain high biological productivity in the subarctic Pacific. Under such warm conditions, the biological pump can be enhanced by the intensification of nutrient-rich subsurface water upwelling, facilitated by reduced sea-ice coverage and elevated ocean surface temperatures[14,36].

## Mechanism driving the nutrient upwelling in the subarctic Pacific Ocean

As biological productivity in the study region is primarily driven by the supply of nutrients via upwelling, we define an upwelling index based on bulk sedimentary $\delta^{15}N$ and productivity proxies (see the Methods for detailed information), following the approach outlined by ref. 14. Similar to productivity proxy of Ba/Ti, primary variations in the upwelling index closely correlate with glacial-interglacial cycles, while secondary variations exhibit a strong correspondence with precession forcing (Fig. 2d; Supplementary Fig. 4). Our reconstructed upwelling index is consistent with a recent study from the Bering Sea[14] (Fig. 3c, d),

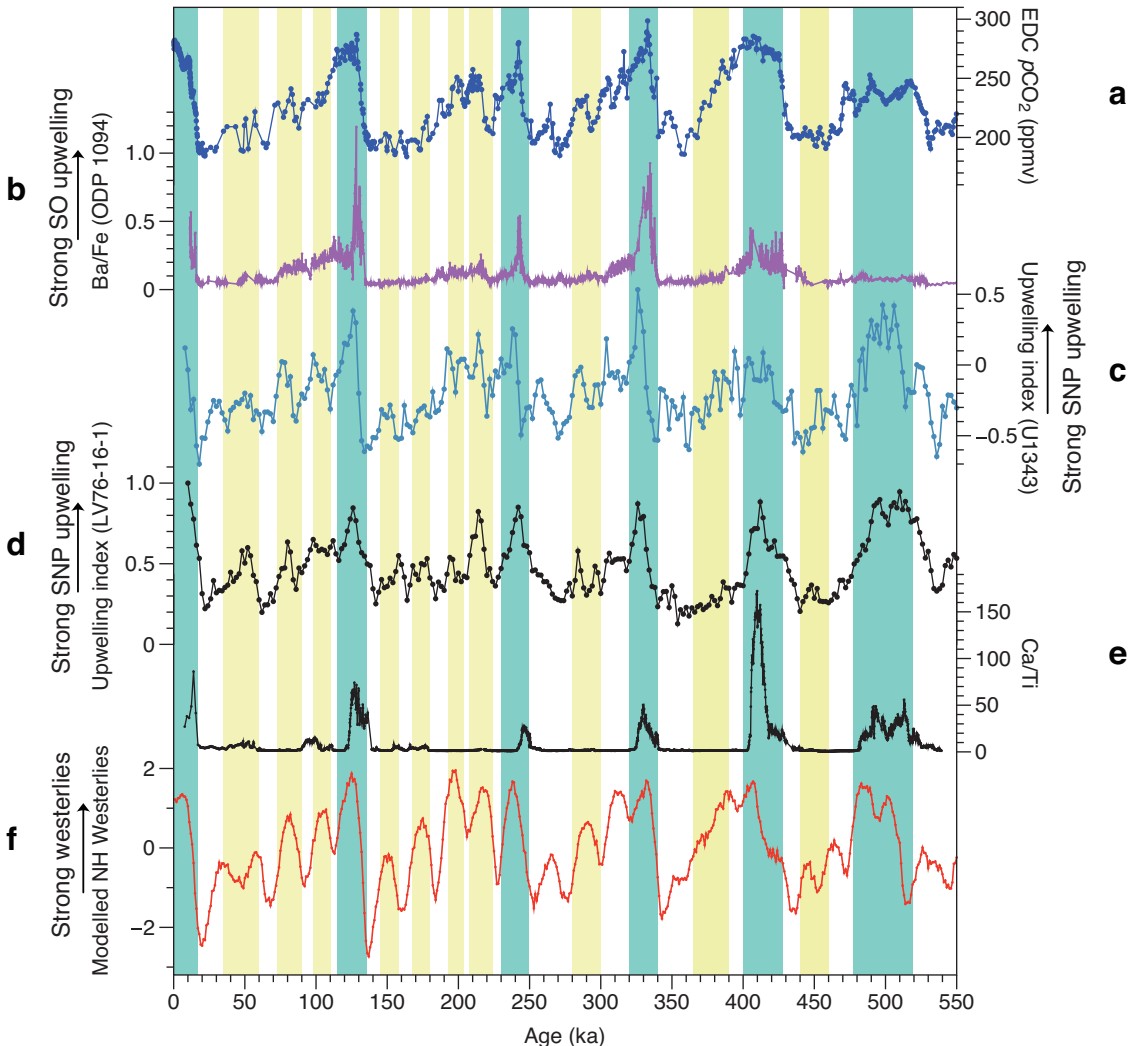

**Fig. 3 | Environmental proxies of core LV76-16-1 and comparison with global and regional climate proxies. a** Atmospheric $CO_2$ concentrations ($pCO_2$) from the Antarctic records[1,66]. **b** Sedimentary Ba/Fe from ODP Site 1094 as a proxy of productivity and upwelling in the Antarctic zone of the Southern Ocean[4]. **c** Upwelling index of IODP Site U1343[14]. **d** Upwelling index of core LV76-16-1. **e** X-ray fluorescence (XRF) scanning Ca/Ti ratio of core LV76-16-1. **f** Simulated annual mean North Pacific westerlies (35°N-55°N, 120°E-240°E, normalized) at 500 hPa (see the Methods for description). The blue bars represent the strongest upwelling which correspond to each major glacial termination. The yellow bars denote the moderate upwelling. Source data are provided as a Source Data file.

revealing that warm interglacial conditions and low precession (equivalent to high boreal summer insolation) conduce to stronger upwelling (Fig. 2d; Supplementary Fig. 4). Similar to the productivity proxies, upwelling during MIS 11 displayed a more pronounced response to obliquity when compared to other interglacial periods.

Both proxies and model simulations[10,37] demonstrate that wind stress plays a key role in controlling Ekman suction and nutrient upwelling in the subarctic Pacific, which is ultimately related to the changes in the position and intensity of the northern westerlies. The northern westerlies are located at ~40° N today, and are the prevailing westerly winds that flow from the subtropical high-pressure belts to the subpolar low-pressure belts in the Northern Hemisphere[37]. These winds resulted from baroclinicity caused by the meridional thermal gradient between the warm subtropical atmosphere and the colder polar atmosphere[38,39], and they are ultimately fueled by the non-uniform distribution of annual solar radiation inputs to the global Earth-atmosphere system[37]. In contrast to the subarctic Pacific area (north of ~40° N), productivity in the subtropical Pacific (south of ~40° N) was higher during glacials, at least over the last 200 kyr[40–43] (Supplementary Fig. 6e–g). We thus propose that the observed seesaw pattern in the export production and nutrient upwelling records spanning the subarctic and subtropical Pacific regions were caused by a meridional shift and/or changes in the strength of the northern westerlies which seem sensitive to changes in glacial conditions. A compilation of planktic foraminiferal $\delta^{18}O$ records from the North Pacific and climate model results indicate a southward migration (~3°) of the westerlies and increased wind stress during the Last Glacial Maximum[18].

We propose that upwelling in the subarctic Pacific was mainly controlled by changes in wind stress and consequently by the variations in the northern westerlies which are controlled by glacial-interglacial ice sheets and orbital forcing. To test our hypothesis and to investigate how the westerlies responded to changes in ice sheets (ICE), greenhouse gases concentrations (GHG), and orbital parameters (Orb), we analyzed the results of three transient simulations performed using the LOVECLIM1.3 model. These simulations cover the period of 133–75 ka, which includes three precession cycles, one-and-a-half obliquity cycle, and large variations in GHG and ICE, and thus are suitable for our research purpose. The first two simulations, which were used and described in ref. 44, consider only the effect of orbital forcing (Orb experiment) and the combined effects of orbital forcing and GHG (OrbGHG experiment), respectively. In the third simulation, the changes in Northern Hemisphere (NH) ice sheets were included (OrbGHGICE experiment), and the experimental setup related to ice sheets was described in ref. 45.

The Orb experiment allows us to investigate the effects of orbital forcing alone. The model outputs show that the intensity of the westerlies and wind stress across the subarctic Pacific show high consistency, and are mainly controlled by precession, with obliquity playing a secondary role (Fig. 4c, d, red line). The model results indicate that low precession (or high boreal summer insolation) leads to stronger westerlies across the subarctic Pacific and by inference stronger upwelling, which is consistent with the abovementioned observation that upwelling was stronger when precession is low. Comparison of two extreme precession cases (i.e., the precession minimum (Pmin) at 127 ka and precession maximum (Pmax) at 97 ka), with similar obliquity, suggests that the westerlies over the northern mid-latitude Pacific were intensified at Pmin (Fig. 4e, g). Further analysis of the model results shows that, compared to Pmax, the low pressure centered over the Kamchatka Peninsula is deepened and the high pressure over the subtropical Pacific is strengthened at Pmin, leading to a larger latitudinal pressure gradient and thus, stronger westerlies.

Comparison between Orb and OrbGHG simulations shows that GHG has little effect on the westerlies over the subarctic Pacific,

compared to the effect of precession. However, the OrbGHGICE simulation shows that the NH ice sheets have a large effect on the intensity and position of the westerlies, an effect which is much larger than that of precession (Fig. 4c, g). The effect of ice sheets (expressed by the difference between OrbGHGICE and OrbGHG) explains a much larger amount of variance of the westerlies in OrbGHGICE (R = 0.90) than the effect of orbital forcing (R = 0.32). The much greater impact of ice sheets can also be observed in Fig. 4g, and by comparing Fig. 4e, f (note that the color scale is different). Figure 4g shows that in response to large NH ice sheets, such as those at 91 ka, not only is the intensity of the westerlies substantially reduced but also its position is shifted southwards. This finding supports the observation that upwelling was weaker during glacials and stronger during interglacials. It is evident that the westerlies respond much less to precession changes, being about three times weaker, and their position is also less affected compared to the response to changes in ice sheets (Fig. 4g). Further analysis shows that large NH ice sheets greatly weaken the low-pressure system centered over the Bering Strait and the surrounding regions (pressure increases) and they slightly weaken the subtropical high-pressure system (pressure decreases), leading to a reduced pressure gradient between the subtropical and subpolar zones, and finally to the weakening and southward shift of the westerlies over the northern Pacific.

Therefore, our model results and proxies consistently show that the westerlies and upwelling over the subarctic Pacific are strongly influenced by ice sheets and precession. Model results also show that the changes in ice sheets exert the primary influence on the westerlies over the subarctic Pacific, with precession playing a secondary role. This aligns with the observation that variations in export productivity and the reconstructed upwelling index primarily follow glacial-interglacial cycles, with precession having a comparatively weaker impact throughout the sequence (Fig. 2a–d; Supplementary Fig. 4). Overall, warm conditions, resulting from a low ice volume and low precession, promote stronger upwelling of nutrient-rich subsurface water.

## Implications for changes in glacial-interglacial atmospheric $p$CO$_2$

The cyclic changes in $p$CO$_2$ on glacial-interglacial timescales are mostly attributed to changes in the upwelling of $CO_2$-rich water in the deep ocean[2,8,10], while the Southern Ocean is considered as the primary control on this $p$CO$_2$ variability[46]. Furthermore, Jaccard et al. (2013) proposed a two-mode mechanism in the Southern Ocean that may control the amplitude and timing of $p$CO$_2$ changes during glacial-interglacial cycles[4]. Peak interglacial pCO$_2$ changes in the Southern Ocean's Antarctic zone were driven by $CO_2$-rich water upwelling, while the transition to glacial conditions involved increased remineralized carbon sequestration in the ocean interior through a stronger Sub-Antarctic zone biological carbon pump aided by iron fertilization[4].

While the Southern Ocean plays a critical role in regulating the Pleistocene $p$CO$_2$ variability, similar processes could occur in the subarctic Pacific[14,46], considering that approximately 30 ppm of $p$CO$_2$ was released into the atmosphere as a result of enhanced overturning in the subarctic region during the last deglaciation[47]. More importantly, the variations in the upwelling index are consistent with those of $p$CO$_2$ and of the modelled northern westerlies since ~550 ka (Fig. 3a, d, f). This suggests that wind-driven upwelling exerts a primary control on changes in $p$CO$_2$. Moreover, the CaCO$_3$ peaks coincided with the strongest upwelling during the warmest interglacials (Fig. 3e) throughout the entire subarctic Pacific[12,13,21]. Besides the effects of increased carbonate production, this reflects abrupt releases of deep-sequestered $CO_2$ from the ocean back into the atmosphere, leading to a higher calcite saturation state in the bottom water[4,12].

The most prominent feature in our productivity and upwelling records is the clear precessional signal in the subarctic Pacific

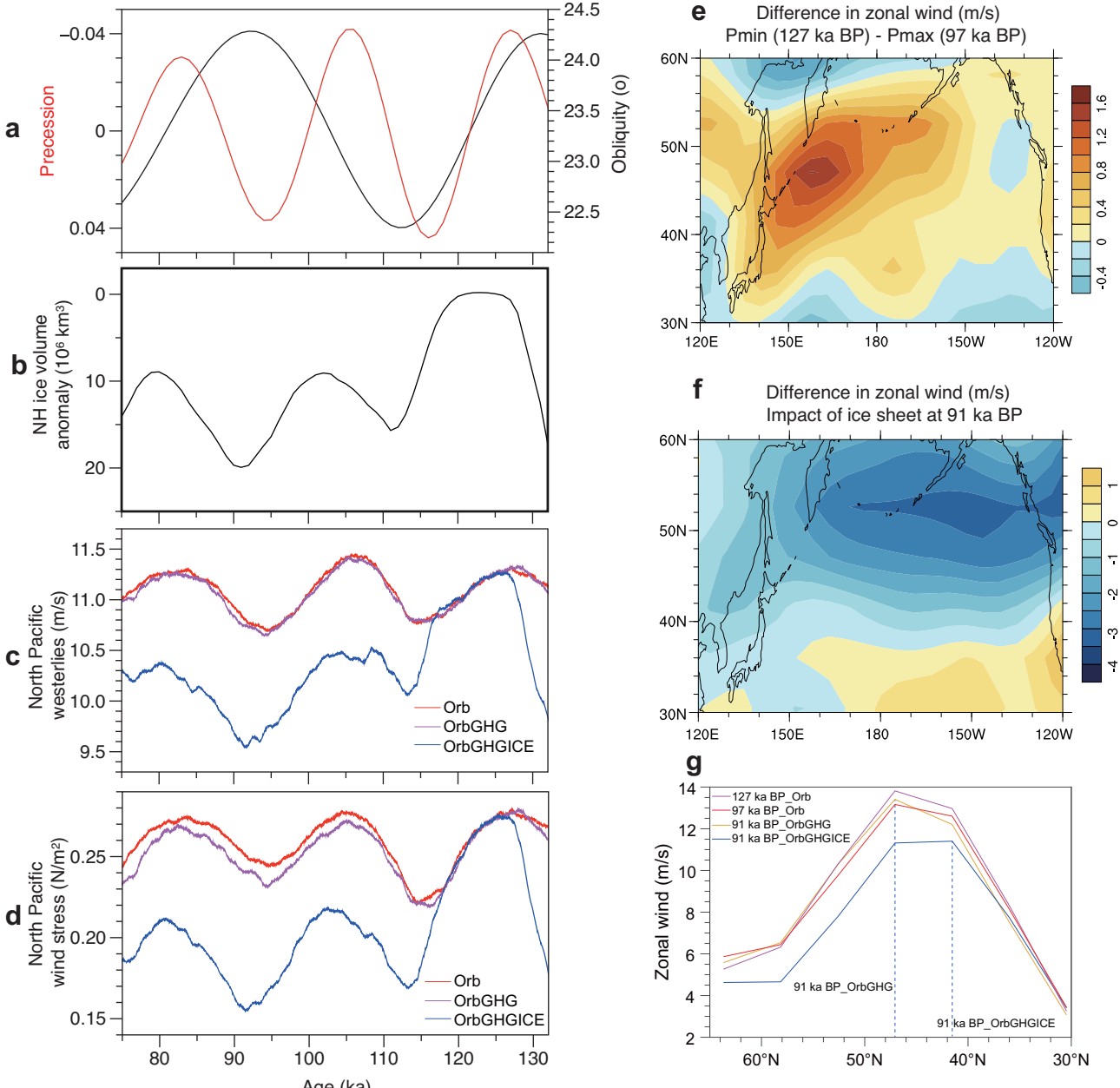

**Fig. 4 | The response of the simulated North Pacific westerlies to orbital forcing, Greenhouse gases (GHG) concentrations, and Northern Hemisphere (NH) ice sheets. a** Precession (red curve) and obliquity (black curve)[60]. **b** Northern Hemisphere (NH) ice volume anomaly as compared to pre-industrial (PI)[64]. **c** Annual mean westerlies at 500 hPa over the North Pacific (35°N-55°N, 120°E-240°E); A 1000-year running mean was applied. **d** Annual mean zonal wind stress over the North Pacific (35°N-55°N, 120°E-240°E); A 1000-year running mean was applied. **e** Difference of the 500-hPa annual mean zonal wind between Pmin (127 ka) and Pmax (97 ka) from the Orb experiment over the North Pacific. **f** Difference of the 500 hPa annual mean zonal wind at 91 ka between the OrbGHGICE and OrbGHG experiments. **g** 500 hPa annual mean zonal wind over the North Pacific for the selected cases in (**e**, **f**). The vertical dashed lines in (**g**) represent the position of the maximum zonal wind. Orb: experiment considering orbital parameters; OrbGHG: experiment considering orbital parameters and greenhouse gases; OrbGHGICE: experiment considering orbital parameters, greenhouse gases and Northern Hemisphere ice sheets. Source data are provided as a Source Data file.

(Fig. 2b–d; Supplementary Fig. 4), which is not evident in the Southern Ocean (Fig. 3b; Supplementary Fig. 7d, e), either in the Antarctic zones[4,48] or the Sub-Antarctic[49], or in the Equatorial Pacific[50]. Most upwelling records inferred from productivity proxies in these regions are dominated by 100 kyr cyclicity with a subsidiary 40 kyr cycle[4,50–52]. Our analysis clearly shows that precession (~20 kyr cycle) plays a more important role in the upwelling in the subarctic Pacific than that in the Southern Ocean. Therefore, the relatively high-frequency (20-kyr) cyclic changes in atmospheric $p$CO$_2$ (Fig. 3a; Supplementary Fig. 4d) could be related to wind-driven upwelling in the subarctic Pacific. A

recent study provides partial support for our inference that CO$_2$ degassing during the second half of the last deglacial occurred in the North Pacific and elsewhere, instead of in the Southern Ocean[15].

In summary, our results shed light on the significant role of the subarctic Pacific in regulating atmospheric $p$CO$_2$ concentrations, primarily through the dynamic interplay of ice sheets and insolation on the northern westerlies and wind-driven upwelling. In particular, we have introduced a mechanism to explain the variations of upwelling in subarctic Pacific, through which differentiate the forcing of ice sheets and insolation. We propose that the subarctic Pacific plays a crucial

role in regulating atmospheric $CO_2$ concentration variations on precession timescale, which would fill a critical gap in our knowledge of the subarctic Pacific's significance in regulating global atmospheric $CO_2$ changes across different timescales. Under warm climatic conditions, the strengthening of the northern westerlies and the enhancement of upwelling in the subarctic Pacific could lead to increased atmospheric $pCO_2$ levels. With the intensification of global warming, the anticipated poleward shift of the northern westerly winds[53] is poised to further amplify the upwelling rate in the subarctic Pacific. This, in turn, would release more $CO_2$ into the atmosphere and further accelerate global warming through positive feedbacks.

## Methods

### Opal and CaCO₃ content analysis

Concentration of biogenic opal of 766 samples (1 cm intervals, ~1 kyr resolution) from core LV76-16-1 was determined by alkaline extraction of silica[54], and was measured by molybdate-blue spectrophotometry. The long-term accuracy of this method is ±0.5 wt%, as deduced from replicate and in-house standard measurements. A total of 383 samples were analyzed at 2 cm intervals (~2 kyr resolution) for measuring $CaCO_3$ content by dissolution in HCl.

### Elemental abundance determination

A total of 383 samples were analyzed for major and trace elements. All samples were first digested with $HNO_3$-HF (1:1) in closed Teflon beakers, and analyzed with ICP-MS (Thermo Scientific X SERIES 2) for trace elements. Quality control were implemented throughout the entire experimental process, including the inclusion of a blank experiment, the use of GSD-9 standard material, and replicate measurements. The relative standard deviations of the major and trace element analyses are all less than 5%.

Concentrations of biogenic Ba (BioBa) can be expected based on the assumption that the composition of Ti in the terrigenous material remained constant in space and time. The elemental ratio from the upper continental crust (UCC)[55] are used for normalization.

$$BioBa = Ba_{sample} - (Ba/Ti)UCC \times Ti_{sample} \qquad (1)$$

### XRF scanning elemental abundance

Elemental abundances (in counts per second; cps) in core LV76-16-1 were measured by X-ray fluorescence (XRF) core scanning at ~5 mm intervals (~0.5 kyr resolution) using the Itrax XRF Core Scanner, with 20 s count times, 30 kV X-ray voltage, and an X-ray current of 40–55 mA.

### Sedimentary δ¹⁵N analysis

A total of 379 bulk samples were measured for nitrogen isotopic analysis on an Isoprime 100 isotopic ratio mass spectrometer (IRMS). Values for $\delta^{15}N$ are reported in per mil notation relative to atmospheric nitrogen gas. A standard sample (protein, $\delta^{15}N = 6.0‰$) was inserted for every 10 samples, and the measured nitrogen isotopes of standards yielded a precision of better than ±0.2‰. The reproducibility of replicate nitrogen isotopic analyses for duplicate samples was generally better than ±0.3‰.

### Ice-rafted debris (IRD) analysis

Ice-rafted debris counts were carried out for 382 samples at ~2 cm interval using a standard reflected light binocular microscope. Each sample weighing approximately 5 g was wet-sieved through a 150 μm sieve. The IRD is defined as the grains (>150 μm) per gram of the dry bulk sediment, as this fraction was regarded as the lower size threshold for IRD categorization[56].

### Calculation of Upwelling index

The upwelling index is calculated following the approach in ref. 14 by using productivity proxy and sedimentary $\delta^{15}N$. Briefly, the normalized sum of $CaCO_3$ and opal (proxy of productivity) were linearly interpolated at an interval of 2 kyr (mean resolution is 1.5 kyr), which was subtracted by the normalized $\Delta\delta^{15}N_{LV76-1012}$ to create a semi-quantitative proxy of the 'upwelling index', with higher values indicating increased nutrient upwelling. The sedimentary $\delta^{15}N$ from ODP Site 1012 in the eastern tropical North Pacific[57] is thought to be a site of nearly complete nitrate utilization throughout the Pleistocene[14,22].

$$\text{Upwelling index} = (CaCO_3 + opal)_{normalized} - (\Delta\delta^{15}N_{LV76-1012})_{normalized} \qquad (2)$$

### Analysis of U-Th isotopes

Uranium and Thorium isotopes of 27 samples were determined by laser ablation Multi-Collector Inductively Coupled Plasma Mass Spectrometry (MC-ICP-MS) on a Neptune plus. Sample preparation and analysis followed the methods described in ref. 58. Briefly, about 0.2–0.3 g of dried sample was homogenized, heated to remove volatile components, packed in a molybdenum capsule and isolated in a graphite tube to avoid oxidation, then heated at 1300 °C for 10–15 min before being quenched in water. By doing this, the samples were transformed into homogeneous glass[58]. Analysis of $^{230}Th$, $^{232}Th$, and $^{238}U$ in glass samples were done by the standard sample bracketing method, on a Neptune plus MC-ICP-MS, coupled with a New Wave 193 Research Laser Ablation system in the State Key Laboratory for Mineral Deposits Research in Nanjing University. The LA-MC-ICPMS was optimized using NIST 612 glass to achieve the best $^{238}U$ and $^{232}Th$ intensities, resulting in a U/Th ratio of around 1.1–1.3. The UO + /U+ in our experiments was less than 1%. The $^{232}Th$ tail at $^{230}Th$ mass was ~120 ppb with RPQ on.

### Estimates of fluxes using ²³⁰Th normalization

The application of $^{230}Th$ normalization allows high-resolution sediment mass flux reconstructions over time which are insensitive to lateral sediment redistribution[19]. The $^{230}Th$-normalized flux of biogenic component (opal, $CaCO_3$, and BioBa; Supplementary Fig. 5) were calculated following the method described in ref. 19:

$$Flux_i = (C_i \times \beta230 \times d)/^{230}Th_{ex0} \qquad (3)$$

Where $Flux_i$ represents the flux of a given constituent i (e.g., opal, $CaCO_3$, BioBa), with a concentration $C_i$ in the sediment deposited at a specific water depth (d). The production rate of $^{230}Th$ ($\beta230 = 2.56 \times 10^5$ dpm/cm³/kyr) was used[59]. $^{230}Th_{ex0}$ (dpm/g) is corrected for radioactive decay since its time of deposition, the fraction supported by uranium within lithogenic material ($^{238}U/^{232}Th = 0.7 \pm 0.1$ in the Pacific Ocean) and the fraction of the in situ $^{230}Th$ produced by decay of authigenic $^{238}U$[19].

### Model and simulations

The model used in this study is LOVECLIM1.3, a three-dimension Earth system Model of Intermediate Complexity, with its atmosphere (ECBilt), ocean and sea ice (CLIO) and terrestrial biosphere (VECODE) components being interactively coupled. The model configuration is identical to that employed in ref. 44 and detailed description can be found therein.

In our study, two types of simulations are used. The first type is transient simulations covering the period of 133–75 ka without acceleration. It includes three simulations. The first two simulations, Orb and OrbGHG, were performed in ref. 44 and detailed description of experimental setup can be found therein. Here we only give some brief introduction. In order to isolate and understand the impact of orbital

forcing, in the Orb simulation, only the change of orbital forcing[60] was taken into account, with the GHG and ice sheets being fixed to their pre-Industrial condition. In the OrbGHG simulation, the change of GHG[61–63] is additionally taken into account. In the third simulation, OrbGHGICE, the change of NH ice sheets[64] was taken into account, but the Southern Hemisphere ice sheets remain fixed to the pre-Industrial condition. The initial conditions were provided by a 2000-year equilibrium experiment with the NH ice sheets, GHG concentrations and astronomical parameters at the starting date of the simulated period. In the presence of land ice, albedo, topography, vegetation and surface soil types corresponding to ice-covered condition were prescribed at corresponding model grids in LOVECLIM1.3. Detailed description of the ice sheet setup can be found in ref. 45.

A transient simulation covering the last 800 ka with 10x acceleration was conducted, with the results of the last 550 ka used. In the transient simulation, variations in orbital forcing and GHG were taken into account, and the ice sheets were fixed to their pre-Industrial condition. To consider the impact of the NH ice sheets on the westerlies throughout the last 550 ka, we apply the relationship between the NH ice volume and the effect of NH ice sheets on the westerlies obtained from the 133–75 ka simulations, which are shown highly and linearly correlated (correlation coefficient R = 0.9), on the ice volume of the last 550 ka to estimate the westerly strength over the sub-arctic Pacific, as shown in Fig. 3f.

## Data availability
Source data are provided with this paper. The data generated in this study have been deposited in the Figshare repository (https://doi.org/10.6084/m9.figshare.25484692).

## Code availability
The code for LOVECLIM1.3 is available at www.climate.be/loveclim.

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

## Acknowledgements

We are grateful to the crew and the shipboard scientific members for collecting the samples. This work was jointly supported by the National Natural Science Foundation of China (42176088, 42130412), the National Key Research and Development Program of China (Grant No. 2023YFF0804600), Laoshan Laboratory project (LSKJ202204200) and the Taishan Scholar Program of Shandong (tspd20181216) that supported the work through funding to Z.Y., X.S. and Y.L. The work on model simulations is supported by the Fonds de la Recherche Scientifique-FNRS (F.R.S.-FNRS) under grants T.0246.23 and T.W019.23. Computational resources have been provided by the supercomputing facilities of the Université catholique de Louvain (CISM/UCL) and the Consortium des Équipements de Calcul Intensif en Fédération Wallonie Bruxelles (CÉCI) funded by F.R.S.-FNRS under convention 2.5020.11. Q.Y. is Research Associate F.R.S.-FNRS. Z.W. is Postdoc Fellow supported by the F.R.S.-FNRS grant T.0246.23. S.G. acknowledges funding by the Russian state budget theme No. 7 AAAA-A17-121021700342-9 of POI FEB RAS and Russian Science Foundation (22-17-00118).

## Author contributions

Z.Y., X.S., Q.Y., and Z.G. designed the study. Y.L., S.G., K.W., J.Z., A.B., Y.V., and Y.Y. helped collect the core materials. Q.Y. and Z.W. generated and analyzed the modeling data. Z.Y., T.C., Q.N., H.W., J.C., and A.Z. collected isotopes and geochemical data. A.W. and G.Y. collected IRD data. Z.Y. and S. J. analyzed proxy data. Z.Y. wrote the first draft of the manuscript with contributions from X.S., Q.Y., S.J., Y.L., Z.G., S.G., K.W., T.C., Z.W., Q.N., J.Z., H.W., J.C., A.W., G.Y., A.Z., A.B., Y.V., and Y.Y. to the final version.

## Competing interests

The authors declare no competing interests.

## Additional information

[1]Key Laboratory of Marine Geology and Metallogeny, Shandong Key Laboratory of Deep-Sea Mineral Resources Development, First Institute of Oceanography, MNR, Qingdao, China. [2]Laboratory for Marine Geology, Qingdao Marine Science and Technology Center, Qingdao, China. [3]Earth and Climate Research Center, Earth and Life Institute, Université catholique de Louvain, Louvain-la-Neuve, Belgium. [4]Institute of Geological Sciences, University of Lausanne, Lausanne, Switzerland. [5]Key Laboratory of Cenozoic Geology and Environment, Institute of Geology and Geophysics, Chinese Academy of Sciences, Beijing, China. [6]V.I. Il'ichev Pacific Oceanological Institute, Far East Branch of Russian Academy of Science, Vladivostok, Russia. [7]State Key Laboratory for Mineral Deposits Research, School of Earth Sciences and Engineering, Nanjing University, Nanjing, China. [8]Key Laboratory of Marine Geology and Environment, Institute of Oceanology, Chinese Academy of Sciences, Qingdao, China. [9]These authors contributed equally: Zhengquan Yao, Xuefa Shi. ✉e-mail: yaozq@fio.org.cn; xfshi@fio.org.cn; qiuzhen.yin@uclouvain.be

