## [Peer Review File · Nature Communications]

Ice sheet and precession controlled subarctic Pacific productivity and upwelling over the last 550,000 yearsReviewers' comments:

Reviewer #1 (Remarks to the Author):

Yao et al. present new high resolution reconstructions of ^{230}Th -normalised export productivity in the subarctic northwest Pacific Ocean over the last 550 kyr derived from opal, calcium carbonate, major and trace elements, sedimentary $\delta^{15}\text{N}$, and ice-rafted debris, using sediment core drilled from the Emperor Seamount chain. While the presentation of the data in the manuscript is overall good, these data were not sufficiently integrated with previously published data in the region, and I found that the major conclusions were not adequately supported by data analysis. I also would like to see better delineation of what is new by the authors, along with better integration of existing relevant data.

At this point, I recommend major revision. In the present form, I do not think that this publication is suitable for Nature Communications. Detailed comments to the authors are provided below, I hope these are helpful for revision.

Title:

I think that ice sheet and precession should not be placed together in the same breath here. Major findings of this work show that ice sheet is the main driver and precession only contribute to a "lesser" extent. Consider revising the title to highlight major and novel findings of this work.

Abstract:

I was confused. The way the Abstract is written seems like this manuscript is about the Southern Ocean. I understand that the authors are trying to compare the study area to the Southern Ocean in terms of carbon reservoir. Consider rewriting the first two sentences to make a better transition to the subarctic Pacific.

L33: Spell-out "NW".

L37: What kind of "biological production"? In L33, you used export productivity.

L38: Delete "somewhat".

Introduction:

Same comment in the Abstract about the way the Southern Ocean is highlighted. Why start with the Southern Ocean? If you do it this way, make sure that you transition the focus to your studied region and discuss the connection of the Southern Ocean to the North Pacific.

The order of the Introduction does not logically flow from large scale picture to focused. We know that most of the studies focus on the role of the Southern Ocean in atmospheric carbon dioxide concentrations during glacial-interglacial cycles. The authors can improve the first paragraph by highlighting the gap in recent studies so that it focuses on the North Pacific. The third paragraph is very important as this is the novel aspect of this work, yet the way the authors introduced what they have done is muted.

L63: Insert "an" before "important".

L71: Replace "their" with "this".

Productivity changes over the past 550 kyr:

L81: Call out Extended Data Fig. 1a first before 1b, c, etc. Please rearrange the figure order.

L85: Replace "with" with "to".

L86: Please see comment in L81.

L87: Replace "between ~1.2-1.7" with "between ~1.2 and 1.7".

L111-113: The precession is not evident in the plot, especially prior to ~250 kyr. Only MIS 5 shows precession. I can see a distinct 100 kyr and 41 kyr cycles in all the records, similar to the LR04 stack.

Since this is highlighted as a significant finding of this research, I suggest running a spectral analysis (simple periodogram, multitaper, and the likes). I would like to see the precession peak in the plot/plots. It would also help doing the wavelet analysis or filter out the precession signal in the Ba/Ti record, to see the temporal evolution of the cyclicities. There could be a shift in orbital frequencies through the 550 kyr record.

L117-118: Add an explanation how astronomical forcing influence the surface waters (change in water mass characteristics) that led to high biological productivity. This section lacks a discussion on oceanography.

L136: Replace "was" with "is".

L138: Add reference/s for "aeolian transport of dust".

L140: You can delete "on one hand".

L141: Which previous studies?

Mechanism driving the nutrient upwelling in the subarctic Pacific Ocean

L152: What do you mean by "first-order variations"?

L153: What do you mean by "second-order variations"?

L153: I am not convinced about the precession signal in the upwelling index. Can you please do a spectral and/or wavelet analysis?

L156: "...responded mainly to obliquity..." Not only during MIS 11 but all through the record. Precession can be observed from ~250 kyr but the amplitude is not comparable to the interglacial periods. Please delete dash in "MIS-11".

L161: Delete the second "s" in "Hemispheres".

L162: Replace "These winds are resulting..." to "These winds resulted from..."

L166: Add "...glacials, at least over the last 200 kyr." because the record is only until around this time.

L198: Replace "Comparisons of..." to "Comparisons between..."

L208-210: Sentence not clear. Please rewrite.

L221: Please mention time interval when precession "exerts a weaker influence". Is it through the 550 kyr record?

Implications for changes in glacial-interglacial atmospheric pCO₂

L224: Please discuss the link between the Southern Ocean and the North Pacific. See comments previous comments on this.

L241: Not convinced about the "clear" precessional signal unless a spectral plot is shown. Look the same in the Southern Ocean, which is dominated by obliquity and/or eccentricity.

L249-256: This manuscript has some significant and novel findings and yet it looks to me that they are not adequately described and claimed. This last paragraph can be better written to highlight these findings.

Methods:

Please mention respective resolution in kyr of sampling interval in each analysis type.

Line 267: Replace "by" with "with".

Line 268-269: Please rephrase "The entire experimental process was under quality control using a blank experiment..."

L276: Delete "the".

L282: Spell-out "N₂".

L292: Add "...the approach in Ref13..."

L294: Delete "/sample".

Line 328-329: Rephrase.

Line 331: Delete "of".

Line 332-333: Rephrase.

Line 334: Add short explanation why "In the Orb simulation, only the change of orbital forcing was taken into account...".

Figures:

Some figure/s need/s to be rearrange based on the order it was called out in the manuscript. Please

see comments above.
Make sure all figures are called out in the manuscript.

Reviewer #2 (Remarks to the Author):

In this paper Yao et al. try to show that the upwelling in the subarctic North Pacific is especially pronounced during warm interglacial periods with high boreal summer insolation. I have to say that I am not convinced by this finding, especially the association between more intense upwelling and high boreal summer insolation.

This sort of analysis has been done before by Galbraith et al. (2008). According to my reading of the Galbraith et al. paper, periods of more intense upwelling in the western North Pacific take place during times with low boreal summer insolation. This is exactly the opposite conclusion that Yao et al. come to in this paper.

Figure 6 in Galbraith et al., in particular, identifies three intervals with low levels of nitrate assimilation during marine isotope stage 5. These should be times of more intense upwelling. The intervals of low nitrate assimilation occur around 72,000, 96,000, and 120,000 years ago, times when the boreal summer insolation was reduced. Yao et al. even develop an upwelling index that is based on the same approach as Galbraith et al., yet they come to the opposite conclusion.

The problem comes down to the age model developed for core LV76-16-1 in relation to the age model developed for core MD2416 in Galbraith et al. (2008). Galbraith et al. at least had a benthic $\delta^{18}\text{O}$ record that they were able to align with LR04. Yao et al. are stuck using $\% \text{CaCO}_3$ and Ca/Ti ratios to align with LR04 instead. I don't see that Yao et al. refer to a benthic $\delta^{18}\text{O}$ record for LV76-16-1 anywhere in their paper. The approach used by Yao et al. is simply not as valid, especially when trying to align a core chronology with the ~ 23 kyr precessional cycle.

To be sure, precessional cycles were not the main focus of the Galbraith et al. paper. But the observations that they published speaks for themselves. Yao et al. have simply drawn the wrong conclusions about what drives the productivity and upwelling in the subarctic Pacific in the precessional band.

Response to Reviewers' comments:

Reviewer #1

Major comments:

Comment 1:

Yao et al. present new high-resolution reconstructions of ^{230}Th -normalised export productivity in the subarctic northwest Pacific Ocean over the last 550 kyr derived from opal, calcium carbonate, major and trace elements, sedimentary $\delta^{15}\text{N}$, and ice-rafted debris, using sediment core drilled from the Emperor Seamount chain. While the presentation of the data in the manuscript is overall good, these data were not sufficiently integrated with previously published data in the region, and I found that the major conclusions were not adequately supported by data analysis. I also would like to see better delineation of what is new by the authors, along with better integration of existing relevant data. At this point, I recommend major revision. In the present form, I do not think that this publication is suitable for Nature Communications. Detailed comments to the authors are provided below, I hope these are helpful for revision.

Reply:

Thanks for the suggestions. We have added more information on the novelty of our study with a better integration of existing relevant data. Please see *Lines 45-59, 70-76 and Lines 268-280* in the revised manuscript. In addition, integration of previous studies has also been made by comparison of our records with other records (see *lines 136-139, 166-169*). A summary of the novelty of our study is given here below.

Upwelling and vertical mixing in the North Pacific play a crucial role on the Pleistocene $p\text{CO}_2$ changes by regulating the release of deeply-sequestered CO_2 into the atmosphere. However, the mechanism driving the upwelling and more generally the role of the subarctic Pacific in regulating Pleistocene $p\text{CO}_2$ variations remain poorly understood. A few studies have specifically addressed the factors controlling the upwelling in the subarctic Pacific, but their records are too short so allow focusing only on the last deglacial transition (e.g., Gray et al., 2018; Du et al., 2018). High-resolution upwelling records spanning multiple glacial cycles in the subarctic Pacific are very limited. There exists so far only one long upwelling record that was obtained from productivity proxies and sedimentary $\delta^{15}\text{N}$ over the past ~800 ka at the IODP Site U1343 in the Bering Sea (Worne et al., 2019). The variations of the reconstructed upwelling index at this site and at our site are highly consistent on orbital timescale (see Fig. 3c, d in revised manuscript), showing the robustness of the upwelling index reconstructed in

both studies. In addition to the long, high-resolution upwelling index, our manuscript also includes the following new aspects:

(1) We have introduced a new mechanism to explain the variations in upwelling and productivity in subarctic Pacific. Our reconstructed upwelling records and climate model results consistently show the crucial role of precession and the Northern Hemisphere ice sheets in regulating the upwelling and export production in subarctic Pacific via their influences on the westerlies. In particular, our model simulations allow to separate and quantify the effect of different forcing factors, which can hardly be done by using proxy records alone.

(2) Our productivity proxy is validated by ^{230}Th normalization, which is critical for interpreting export production in marine sediments (Francois et al., 2004) and thus upwelling. Such validated records spanning multiple glacial cycles are rare in the subarctic Pacific region.

(3) We propose that the subarctic Pacific plays a crucial role in regulating atmospheric CO_2 concentration variations on precession timescale, which is completely original and has not been addressed in other study. This finding would fill a critical gap in our knowledge of the subarctic Pacific's significance in regulating global atmospheric CO_2 changes across different timescales.

Comment 2:

Title:

I think that ice sheet and precession should not be placed together in the same breath here. Major findings of this work show that ice sheet is the main driver and precession only contribute to a "lesser" extent. Consider revising the title to highlight major and novel findings of this work.

Reply:

Thank you for the suggestion. Indeed, both our proxy records and model simulations show that ice sheet is the primary driver and precession is secondary. However, both drivers are needed in explaining the variability of the productivity and upwelling in the subarctic Pacific, with ice sheet contributing mainly to the variability of the 100-kyr glacial cycles and precession to the variability of ~20-kyr cycles. Moreover, the clear precession signal in our records, which is distinctly different from the Southern Ocean records, is one of the original findings in our study. Therefore, we would keep precession in the title, but slightly change the title to “Ice sheet and precession control on biological export production and upwelling in the subarctic Pacific over the last 550,000 years”.

Comment 3:

Abstract:

I was confused. The way the Abstract is written seems like this manuscript is about the Southern Ocean. I understand that the authors are trying to compare the study area to the Southern Ocean in terms of carbon reservoir. Consider rewriting the first two sentences to make a better transition to the subarctic Pacific.

Reply: Thanks for the suggestion. We agree with the reviewer and have rewritten the first two sentences in the Abstract (*Lines 29-31*) as the following:

“The polar oceans play a vital role in regulating atmospheric CO₂ concentrations during the Pleistocene glacial cycles. However, despite being the largest modern reservoir of respired carbon, the impact of the subarctic Pacific remains poorly understood due to limited records.”

Comment 4:

Introduction:

Same comment in the Abstract about the way the Southern Ocean is highlighted. Why start with the Southern Ocean? If you do it this way, make sure that you transition the focus to your studied region and discuss the connection of the Southern Ocean to the North Pacific.

The order of the Introduction does not logically flow from large scale picture to focused. We know that most of the studies focus on the role of the Southern Ocean in atmospheric carbon dioxide concentrations during glacial-interglacial cycles. The authors can improve the first paragraph by highlighting the gap in recent studies so that it focuses on the North Pacific. The third paragraph is very important as this is the novel aspect of this work, yet the way the authors introduced what they have done is muted.

Reply: As suggested by reviewer 1, we have rewritten the first and third paragraphs in the Introduction section. Please see *Lines 45-59 and 70-76*.

Comment 5:

L111-113: The precession is not evident in the plot, especially prior to ~250 kyr. Only MIS 5 shows precession. I can see a distinct 100 kyr and 41 kyr cycles in all the records, similar to the LR04 stack. Since this is highlighted as a significant finding of this research, I suggest running a spectral analysis (simple periodogram, multitaper, and the likes). I would like to see the precession peak in the plot/plots. It would also help doing the wavelet analysis or filter out the precession signal in the Ba/Ti record, to see the temporal evolution of the cyclicities. There could be a shift in orbital frequencies through the 550 kyr record.

L153: I am not convinced about the precession signal in the upwelling index. Can you please

do a spectral and/or wavelet analysis?

L241: Not convinced about the “clear” precessional signal unless a spectral plot is shown. Look the same in the Southern Ocean, which is dominated by obliquity and/or eccentricity.

Reply:

We thank the reviewer for the suggestion, and agree that the suggested analysis would make our point more convincing. In our revised manuscript, we have added two figures as supplementary materials (Extended Data Fig. 4 and Extended Data Fig. 7, also given below) to show the relevant analysis we have added and to strengthen our conclusion.

Extended Data Fig. 4a clearly shows that the precessional signal filtered out from our Ba/Ti record compares very well with precession throughout the last 550 kyr. The wavelet analysis of our Ba/Ti and upwelling records (Extended Data Fig. 4b, c) shows three major periodicities of ~100 ka, ~40 ka and ~20 ka. It is worth noting that the power of the ~20 ka cycle (so does the power of the ~40 ka cycle) is varying in time mainly due to the amplitude modulation of precession by eccentricity. However, what is new and exciting in our study is that the power of the ~20 ka periodicity in our proxies is equivalent to or even stronger than the power of the 40-ka one over most time of the last 550 ka. This is distinctly different from the productivity records (used to indicate upwelling) in the Southern Ocean (Extended Data Fig. 7d, e) in which the precession signal is hardly visible, but it is clearly visible even in the curves of our raw data (Extended Data Fig. 7a-c). Our analysis clearly show that precession (~20 ka cycle) plays a more important role in the upwelling in the subarctic Pacific than that in the Southern Ocean, which further suggests that the upwelling in the subarctic Pacific might have contributed to the ~20 ka cycle in the variations of the atmospheric CO₂ concentration.

Relevant discussions have been added in the revised manuscript (*Lines 113-123, 164-166, 257-261*).

Extended Data Fig. 4. Variations in Ba/Ti, wavelet analysis of Ba/Ti and upwelling index from core LV76-16-1. (a) Comparison between Ba/Ti (dark line), 20-kyr filtering (blue line), Precession (red line) and Eccentricity (green line) (Berger and Loutre, 1991). (b) Wavelet analysis of Ba/Ti from core LV76-16-1. (c) Wavelet analysis of upwelling index from core LV76-16-1.

Extended Data Fig. 7. Comparison between productivity, upwelling proxies of Core LV76-16-1 and records from the Southern Ocean. (a) Variations in upwelling index of LV76-16-1. (b) Variations in Ba/Ti of LV76-16-1. (c) Variations in opal of LV76-16-1. (d) Variations in opal of Core RC13-259 from the Southern Ocean (Charles et al., 1991). (e) Variations in Ba/Fe of ODP Site 1094 from the Southern Ocean (Jaccard et al., 2013).

Comment 6:

L117-118: Add an explanation how astronomical forcing influence the surface waters (change in water mass characteristics) that led to high biological productivity. This section lacks a discussion on oceanography.

Reply:

An explanation regarding this issue has been incorporated into the revised manuscript (*Lines 155-159*).

“Our results suggest that warm interglacial conditions and high boreal summer insolation, resulting from high obliquity and low precession, conspire to sustain high biological productivity in the subarctic Pacific. Under such warm conditions, the biological pump can be enhanced by the intensification of nutrient-rich subsurface water upwelling, facilitated by reduced sea-ice coverage and elevated ocean surface temperatures (Worne et al., 2019; Kender et al., 2018).”

Comment 7:

L156: “...responded mainly to obliquity...” Not only during MIS 11 but all through the record. Precession can be observed from ~250 kyr but the amplitude is not comparable to the interglacial periods.

Reply:

Please see our reply to Comment 5. The precession signal is clear throughout the 550 kyr although its power is varying in time due to the amplitude modulation by eccentricity. The power of the ~20 kyr cycle is equivalent to or even stronger than the ~40 kyr cycle over most time of the 550 kyr (Extended Data Fig. 4).

Comment 8:

L224: Please discuss the link between the Southern Ocean and the North Pacific. See comments previous comments on this.

Reply:

We have included relevant information regarding the link between the Southern Ocean and the North Pacific (*Lines 247-250*).

“While the Southern Ocean plays a pivotal role in regulating the Pleistocene $p\text{CO}_2$ variability, similar processes could occur in the subarctic Pacific (Worne et al., 2019; Sigman et al., 2010), considering that approximately 30 ppm of $p\text{CO}_2$ was released into the atmosphere as a result of enhanced overturning in the subarctic region during the last deglaciation (Rae et al., 2014).”

Comment 9:

L249-256: This manuscript has some significant and novel findings and yet it looks to me that they are not adequately described and claimed. This last paragraph can be better written to highlight these findings.

Reply:

As recommended by the reviewer, the final paragraph has been revised to elucidate the significance of these findings (*Line 268-280*).

“In summary, our results shed light on the significant role of the subarctic Pacific in regulating atmospheric $p\text{CO}_2$ concentrations, primarily through the dynamic interplay of ice sheets and insolation on the northern westerlies and wind-driven upwelling. In particular, we have introduced a new mechanism to explain the variations of upwelling in subarctic Pacific, through which differentiate the forcing of ice sheets and insolation. We propose that the subarctic Pacific plays a crucial role in regulating atmospheric CO_2 concentration variations on precession timescale, which would fill a critical gap in our knowledge of the subarctic Pacific's significance in regulating global atmospheric CO_2 changes across different timescales. Under warm climatic conditions, the strengthening of the northern westerlies and the enhancement of upwelling in the subarctic Pacific could lead to increased atmospheric $p\text{CO}_2$ levels. With the intensification of global warming, the anticipated poleward shift of the northern westerly winds (Yang et al., 2020) is poised to further amplify the upwelling rate in the subarctic Pacific. This, in turn, would release more CO_2 into the atmosphere and further accelerate global warming through positive feedbacks.”

Other Comments:

L33: Spell-out “NW”.

Reply: Done.

L37: What kind of “biological production”? In L33, you used export productivity.

Reply: We have changed “biological production” to “export productivity” to maintain consistency.

L38: Delete “somewhat”.

Reply: Done.

L63: Insert “an” before “important”.

Reply: Done.

L71: Replace “their” with “this”.

Reply: Done.

L81: Call out Extended Data Fig. 1a first before 1b, c, etc. Please rearrange the figure order.

Reply: Done.

L85: Replace “with” with “to”.

Reply: Done.

L86: Please see comment in L81.

Reply: Done.

L87: Replace “between ~1.2-1.7” with “between ~1.2 and 1.7”.

Reply: Done.

L136: Replace “was” with “is”.

Reply: Done.

L138: Add reference/s for “aeolian transport of dust”.

Reply: Done.

L140: You can delete “on one hand”.

Reply: Done.

L141: Which previous studies?

Reply: We have revised this sentence to enhance clarity of meaning. “On the other hand, previous study suggested that Fe input associated with increased dust supply during the glacials did not exert a substantial influence on export production (Kohfeld and Chase, 2011).” Please see *Lines 148-150*.

L152: What do you mean by “first-order variations”?

Reply: We have replaced this phrase by “primary variations” to enhance clarity of meaning.

L153: What do you mean by “second-order variations”?

Reply: We have replaced this phrase by “secondary variations” to enhance clarity of meaning.

L161: Delete the second “s” in “Hemispheres”.

Reply: Done.

L162: Replace “These winds are resulting...” to “These winds resulted from...”

Reply: Done.

L166: Add “...glacials, at least over the last 200 kyr.” because the record is only until around this time.

Reply: Done.

L198: Replace “Comparisons of...” to “Comparisons between...”

Reply: Done.

L208-210: Sentence not clear. Please rewrite.

Reply: This sentence has been changed to “It is evident that the westerlies respond much less to precession changes, being about three times weaker, and their position is also less affected compared to the response to changes in ice sheets.” Please see *Lines 222-225*.

L221: Please mention time interval when precession “exerts a weaker influence”. Is it through the 550 kyr record?

Reply: Please see our reply to Comment 5. The precession signal is clear throughout the 550 kyr although its power is varying in time due to the amplitude modulation by eccentricity. Thus, we rephrased this sentence. “This aligns with the observation that variations in export productivity and the reconstructed upwelling index primarily follow glacial-interglacial cycles, with precession having a comparatively weaker impact throughout the sequence.” Please see *Lines 233-236*.

Please mention respective resolution in kyr of sampling interval in each analysis type.

Reply: Done.

Line 267: Replace “by” with “with”.

Reply: Done.

Line 268-269: Please rephrase “The entire experimental process was under quality control using a blank experiment...”

Reply: This sentence has been changed to “Quality control were implemented throughout the entire experimental process, including the inclusion of a blank experiment, the use of GSD-9 standard material, and replicate measurements.”

L276: Delete “the”.

Reply: Done.

L282: Spell-out “N₂”.

Reply: Done.

L292: Add “...the approach in Ref¹³...”

Reply: Done.

L294: Delete “/sample”.

Reply: Done.

Line 328-329: Rephrase.

Reply: This sentence has been changed to “The model configuration is identical to that employed in ref. (Yin et al., 2021) and detailed description can be found therein.”

Line 331: Delete “of”.

Reply: Done.

Line 332-333: Rephrase.

Reply: This sentence has been changed to “The first two simulations, Orb and OrbGHG, were performed in ref. (Yin et al., 2021) and detailed description of experimental setup can be found therein.”

Line 334: Add short explanation why “In the Orb simulation, only the change of orbital forcing was taken into account...”.

Reply: This sentence has been changed to “In order to isolate and understand the impact of orbital forcing, in the Orb simulation, only the change of orbital forcing (Berger and Loutre, 1991) was taken into account, with the GHG and ice sheets being fixed to their pre-Industrial condition.”

Some figure/s need/s to be rearrange based on the order it was called out in the manuscript. Please see comments above.

Make sure all figures are called out in the manuscript.

Reply: Done.

Reviewer #2

Comment 1:

In this paper Yao et al. try to show that the upwelling in the subarctic North Pacific is especially pronounced during warm interglacial periods with high boreal summer insolation. I have to say that I am not convinced by this finding, especially the association between more intense upwelling and high boreal summer insolation.

This sort of analysis has been done before by Galbraith et al. (2008). According to my reading of the Galbraith et al. paper, periods of more intense upwelling in the western North Pacific take place during times with low boreal summer insolation. This is exactly the opposite conclusion that Yao et al. come to in this paper.

Figure 6 in Galbraith et al., in particular, identifies three intervals with low levels of nitrate assimilation during marine isotope stage 5. These should be times of more intense upwelling. The intervals of low nitrate assimilation occur around 72,000, 96,000, and 120,000 years ago, times when the boreal summer insolation was reduced. Yao et al. even develop an upwelling index that is based on the same approach as Galbraith et al., yet they come to the opposite conclusion.

Reply:

We thank the reviewer for the comments, and appreciate to have the opportunity to clarify on these points.

After a thorough examination of the Galbraith et al. (2008) paper and a comparison of their data with insolation (which was not conducted in their paper), it appears that there is an inaccurate interpretation in the reviewer's assessment of Figure 6 in Galbraith et al. (2008). The figure below is a plot that we have made to compare directly the record of Galbraith et al. (2008) (a) with summer insolation (b), our upwelling index (c) and an upwelling index in the Bering Sea (d). It shows that the peaks of strong upwelling (indicated by low $\delta^{15}\text{N}$ values) in Galbraith et al. (2008) during MIS-5e and MIS-5c align well with summer insolation peaks. The peak of MIS-5a lags insolation peak by ~5 kyr, possibly related to chronological uncertainties. In our record (c) and the one from Bering Sea (d), the three strong upwelling peaks during MIS-5a, -5b and -5c all correspond well with high summer insolation. Therefore, all these three records support our conclusion that more intense upwelling is associated with high boreal summer insolation. This relationship between upwelling and insolation can be extended to older periods of the past 600 kyr, as can be observed in Fig. 2d and Fig. 3c, d in the revised manuscript. Moreover, this relationship is confirmed by our model results as shown in our manuscript.

Comparison of upwelling index of core LV76-16-1 with other upwelling and climate records. (a) Sedimentary $\delta^{15}\text{N}$ in core MD2416 in the NW Pacific (Galbraith et al., 2008). **(b)** Summer insolation at 65° in the northern Hemisphere (Berger and Loutre, 1991). **(c)** Reconstructed upwelling index of core LV76-16-1 (this study). **(d)** Upwelling index at IODP Site U1343 in the Bering Sea (Worne et al., 2019). The vertical dashed lines represent peaks of boreal summer insolation.

Comment 2:

The problem comes down to the age model developed for core LV76-16-1 in relation to the age model developed for core MD2416 in Galbraith et al. (2008). Galbraith et al. at least had a benthic $d^{18}\text{O}$ record that they were able to align with LR04. Yao et al. are stuck using % CaCO_3 and Ca/Ti ratios to align with LR04 instead. I don't see that Yao et al. refer to a benthic $d^{18}\text{O}$ record for LV76-16-1 anywhere in their paper. The approach used by Yao et al. is simply not

as valid, especially when trying to align a core chronology with the ~23 kyr precessional cycle.

Reply:

As explained in our reply to Comment 1 of Reviewer 2, the problem claimed by the reviewer on the relationship between upwelling and insolation actually does not exist. Nevertheless, here we would like to provide more clarification and evidence to demonstrate the robustness of our age model.

Our age model was established by associating abrupt increases in the Ca/Ti ratios with glacial terminations. The correlation is grounded in the observation that each deglacial transition is marked by abrupt increases in the sedimentary accumulation of biogenic carbonate across the entire North Pacific region (Jaccard et al., 2005; Kohfeld and Chase, 2011). This method of age model construction has been widely employed in the subarctic Pacific, as demonstrated at sites such as ODP Site 882 (Jaccard et al., 2005) and core MD2416 (Galbraith et al., 2008). Furthermore, the downcore variations in the Ca/Ti ratio of core LV76-16-1 in our study closely resemble the variations in the Ca/Al ratio and Ca counts at the neighboring ODP Site 882 and core MD2416 (Extended Data Fig. 2), both of which have well-established chronology ((Jaccard et al., 2005; Galbraith et al., 2008). The consistency observed across different cores confirms the robustness of our age model and underscores its suitability for addressing the issues raised in our manuscript.

Extended Data Fig. 2. Comparison of Ca/Ti in core LV76-16-1 with other records. (a) Ca/Ti ratio of core LV76-16-1 (this study). **(b)** XRF scanning Ca counts in core MD2416 in the NW Pacific (Galbraith et al., 2008). **(c)** Ca/Al ratio at ODP Site 882 in the NW Pacific (Jaccard et al., 2005). **(d)** The LR04 benthic $\delta^{18}\text{O}$ stack (Lisiecki and Raymo, 2005). The vertical yellow bars represent interglacial periods.

References

- Berger, A. & Loutre, M-F. Insolation values for the climate of the last 10 million of years. *Quat. Sci. Rev.* **10** (4), 297–317 (1991).
- Charles, C. D., Froelich, P. N., Zibello, M. A., Mortlock, R. A. & Morley, J. J. Biogenic opal in Southern Ocean sediments over the last 450,000 years: implications for surface water chemistry and circulation. *Paleoceanography* **6**, 697–728 (1991).
- Du, J., Haley, B. A., Mix, A. C., Walczak, M. H. & Praetorius, S. K. Flushing of the deep Pacific

- Ocean and the deglacial rise of atmospheric CO₂ concentrations. *Nat. Geosci.* **11**, 749–755(2018).
- Francois, R. M., Frank, M. M. R., van der Loeff & Bacon, M. P. Th normalization: An essential tool for interpreting sedimentary fluxes during the late Quaternary. *Paleoceanography* **19**, PA1018 (2004).
- Galbraith, E. D. et al. Consistent relationship between global climate and surface nitrate utilization in the western subarctic Pacific throughout the last 500 ka. *Paleoceanography* **23**, 1–11 (2008).
- Gray, W. R. et al. Deglacial upwelling, productivity and CO₂ outgassing in the North Pacific Ocean. *Nat. Geosci.* **11**, 1 (2018).
- Jaccard, S. L. et al. Glacial/interglacial changes in subantarctic North Pacific stratification. *Science* **308**, 1003–1005 (2005).
- Jaccard, S. L. et al. Two modes of change in Southern Ocean productivity over the past million years. *Science* **339**, 1419–1423 (2013).
- Kender, S. et al. Closure of the Bering strait caused mid-Pleistocene transition cooling. *Nat. Commun.* **9** (2018).
- Kohfeld, K. E. & Chase, Z. Controls on deglacial changes in biogenic fluxes in the North Pacific Ocean. *Quat. Sci. Rev.* **30**, 3350–3363 (2011).
- Lisiecki, L. E. & Raymo, M. E. A Pliocene-Pleistocene stack of 57 globally distributed benthic $\delta^{18}\text{O}$ records. *Paleoceanography* **20**, 1–17 (2005).
- Rae, J.W.B. et al. Deep water formation in the North Pacific and deglacial CO₂ rise. *Paleoceanography* **29**, 1–23 (2014).
- Sigman, D.M., Hain, M.P. & Haug, G.H. The polar ocean and glacial cycles in atmospheric CO₂ concentration. *Nature* **466**, 47–55 (2010).
- Worne, S., Kender, S., Swann, G. E. A., Leng, M. J. & Ravelo, A. C. Coupled climate and subarctic Pacific nutrient upwelling over the last 850,000 years. *Earth Planet. Sci. Lett.* **522**, 87–97 (2019).
- Yang, H. et al., Poleward shift of the major ocean gyres detected in a warming climate. *Geophys. Res. Lett.* **47**, e2019GL085868 (2020).
- Yin Q. Z., Wu Z. P., Berger A., Goosse H. & Hodell D. Insolation triggered abrupt weakening of Atlantic circulation at the end of interglacials. *Science* **373**, 1035–1040 (2021).

REVIEWER COMMENTS

Reviewer #1 (Remarks to the Author):

Yao et al. have substantially revised their manuscript and addressed all my comments, especially my questions on cyclicities.

The manuscript is much more organized, readable and clear than the first version, and excellent in sections. With that said, I am accepting this version as ready for publication.

I want to emphasize that this manuscript covers a lot of interesting and new grounds at the intersection of biological, physical, and climate changes of the recent glacial cycles in the Pacific, and I believe this should be published.

Reviewer #2 (Remarks to the Author):

The main complaint in my original review was about Yao et al.'s insistence that the strongest upwelling in the subarctic North Pacific occurs during maxima in the northern summer insolation. I identified work by Galbraith et al. (2008) that seemed to show the opposite tendency, i.e., stronger upwelling during minima in the northern summer insolation. I would agree that the upwelling in the subarctic North Pacific varies with the precessional forcing, but it is far from certain that the strongest upwelling occurs during the insolation maxima.

Maxima and minima in the precessional forcing are separated by only 10-11,000 years. The chronologies of the cores in question, LV76-16-1 in this paper and MD2416 in the Galbraith et al. paper, are constrained by spikes in the Ca or Ca/Ti contents that occur every 100,000 years or so during the big deglaciations. The chronologies at all other times are based on an assumption that the sedimentation between the spikes is constant over time. It would not take much variability in the sedimentation between the deglaciations to obscure 10-11,000-yr differences in the timing of events.

A third record has since come to my attention in Yao et al.'s response to Reviewer 1. This record is from IODP site U1343 in the Bering Sea, as described in Worne et al. (2019). Worne et al. see evidence of upwelling variations that are similar to the variations seen at the other sites. The sedimentation rate at site U1343 is higher than at the other locations, however, and the sediments were recovered from a shallower depth. The sediments also contain benthic forams from which a $\delta^{18}O$ record was constructed. This is important because Worne et al.'s $\delta^{18}O$ record provides time markers for the intervals between the deglaciations.

According to the upwelling index in the red curve in Worne et al.'s Figure S3, the strongest upwelling during stages 5 and 7 in the Bering Sea occurred around 70, 95, 190, and 210 ka. As I was inclined to think in my original view, the strongest upwelling at site U1343 occurred when the northern summer insolation was relatively weak. Yao et al. use the same upwelling index, yet their results seem completely different. The differences, it seems to me, arise from the chronologies of the two cores. One is better resolved than the other.

Yao et al.'s entire paper is built around their certainty about the upwelling variations at site LV76-16-1. I don't see any justification for this level of certainty. My recommendation at this point is that the paper should be rejected.

Reviewer #3 (Remarks to the Author):

This is a comment to the manuscript by Yao et al. and specifically the reviewer's comments, which

show some opposite opinions after two rounds of review.

Reviewer 1 provides an extensive and constructive review, which is was answered in detail by Yao et al. Reviewer 2 has a problem with the age model presented by Yao et al. and argues that the paper by Galbraith et al. shows opposite results because of the age model problem. However, Yao et al. argue that the interpretation of the reviewer 2 was not correct and provide an explanation that appears valid.

What additionally argues for me that the age model, and therewith the story, is valid, is that their age model and trends in productivity on themselves are not something new for the north Pacific. These productivity peaks have been identified at many locations following a similar mechanism, including reference to the Galbraith et al. paper. This is also where their age model is based on, i.e. comparing these peaks to the neighboring ODP Site 882 and align them to construct the age model.

Obviously, and also determined by setting, the age model would not be valid for high-resolution conclusions on leads and lags during the termination. But Yao et al. are also staying away from this. One, as a minor side comment, thing that I noticed in the age model plot was that the variations in sedimentation rate are not very large due to the limited number of tie-points. Although difficult to constrain, these peaks in productivity probably increased sedimentation rates significantly during the time intervals that these conditions prevailed. But this would not change the correlation of the peaks with those at other locations in the north Pacific as these would have experienced the same changes.

As an important part of the paper, it is concluded that the precession has a significant impact on the cyclicity of the productivity and thus on CO₂ release. What would be interesting to include in figure showing atmospheric CO₂ is if this actually also contains the precession component. If the reservoir in the north Pacific is so large, this should be present in the overall signal as well.

Reply to the reviewers' comments on "Ice sheet and precession control on biological export production and upwelling in the subarctic Pacific over the last 550,000 years" by Yao et al.

Reviewer #1

Comment:

Yao et al. have substantially revised their manuscript and addressed all my comments, especially my questions on cyclicities. The manuscript is much more organized, readable and clear than the first version, and excellent in sections. With that said, I am accepting this version as ready for publication. I want to emphasize that this manuscript covers a lot of interesting and new grounds at the intersection of biological, physical, and climate changes of the recent glacial cycles in the Pacific, and I believe this should be published.

Reply:

We sincerely appreciate your thoughtful and constructive comments during the initial review, which has significantly enhanced the quality of our manuscript. Your positive comments and the confirmation of acceptance for publication greatly encourage us.

Reviewer #2

Comment:

The main complaint in my original review was about Yao et al.'s insistence that the strongest upwelling in the subarctic North Pacific occurs during maxima in the northern summer insolation. I identified work by Galbraith et al. (2008) that seemed to show the opposite tendency, i.e., stronger upwelling during minima in the northern summer insolation. I would agree that the upwelling in the subarctic North Pacific varies with the precessional forcing, but it is far from certain that the strongest upwelling occurs during the insolation maxima.

Maxima and minima in the precessional forcing are separated by only 10-11,000 years. The chronologies of the cores in question, LV76-16-1 in this paper and MD2416 in the Galbraith et al. paper, are constrained by spikes in the Ca or Ca/Ti contents that occur every 100,000 years or so during the big deglaciations. The chronologies at all other times are based on an assumption that the sedimentation between the spikes is constant over time. It would not take much variability in the sedimentation between the deglaciations to obscure 10-11,000-yr

differences in the timing of events.

A third record has since come to my attention in Yao et al.'s response to Reviewer 1. This record is from IODP site U1343 in the Bering Sea, as described in Worne et al. (2019). Worne et al. see evidence of upwelling variations that are similar to the variations seen at the other sites. The sedimentation rate at site U1343 is higher than at the other locations, however, and the sediments were recovered from a shallower depth. The sediments also contain benthic forams from which a $d^{18}O$ record was constructed. This is important because Worne et al.'s $d^{18}O$ record provides time markers for the intervals between the deglaciations.

According to the upwelling index in the red curve in Worne et al.'s Figure S3, the strongest upwelling during stages 5 and 7 in the Bering Sea occurred around 70, 95, 190, and 210 ka. As I was inclined to think in my original view, the strongest upwelling at site U1343 occurred when the northern summer insolation was relatively weak. Yao et al. use the same upwelling index, yet their results seem completely different. The differences, it seems to me, arise from the chronologies of the two cores. One is better resolved than the other.

Yao et al.'s entire paper is built around their certainty about the upwelling variations at site LV76-16-1. I don't see any justification for this level of certainty. My recommendation at this point is that the paper should be rejected.

Reply:

We would like to extend our gratitude for your patience and for undertaking another review of our manuscript. Following your comments, we have implemented the following revisions and clarifications.

Chronology framework: To address concerns regarding the age model of our study core (LV76-16-1), we have enriched our manuscript with additional details to reinforce the credibility of our age model. The chronology for ODP Site 882 was established by matching the high-resolution XRF scanning Ba/Al ratios with the millennial-suborbital variations of δD in the Antarctic ice core (Jaccard et al., 2005; Galbraith et al., 2008). For core MD2416, the age model was developed by aligning XRF scanning Ca counts with those from ODP Site 882. This methodology underscores that the age model for core LV76-16-1 has a high enough resolution to allow for discussions of suborbital-scale variations. While we recognize that there is some degree of uncertainty in our chronology and that of others, the consistent trends in productivity and other parameters across numerous sites in the North Pacific lend strong support to the robustness of our age model. This information has been added in the revised manuscript (**P4, Lines 91-96**).

“Specifically, the chronology for ODP Site 882 was established by correlating high-

resolution XRF scanning Ba/Al ratios with the millennial-suborbital variability of δD from Antarctica ice core^{12,22}. The age model for MD2416 were developed by aligning XRF scanning Ca counts with those from ODP Site 882²². This demonstrates that the chronology of core LV76-16-1 has a high enough resolution to allow for discussions of suborbital-scale variations.”

Concerns on the relationship between upwelling intensity and insolation: In light of the feedback regarding the upwelling index presented by Worne et al., 2019, Reviewer #2 claimed that the most significant upwelling events in the Bering Sea during stages 5 and 7 occurred around 70, 95, 190, and 210 ka, coinciding with periods of relatively low northern summer insolation. We believe this interpretation might have been an oversight. To clarify, we have produced a plot comparing Worne's upwelling index with Northern Hemisphere summer insolation, which clearly shows that strong upwelling phases mostly align with periods of strong northern summer insolation (refer to Supplementary Figure S1 shown below). This pattern is in agreement with observations from our study location, LV76-16-1. While there are slight discrepancies in certain periods, likely attributable to chronological precision, these do not detract from the fundamental relationship observed. In our manuscript, we have already illustrated the comparison between the upwelling index at LV76-16-1 and northern summer insolation (as seen in Figure 2d), alongside the correlation between Worne's U1343 site and LV76-16-1 (presented in Figures 3c and d). To maintain the manuscript's conciseness and avoid redundancy, we have chosen not to include Supplementary Figure S1 in the main text.

In conclusion, our findings and the methodological approach to establishing a chronological framework are robust, providing a sound basis for discussing upwelling variations in relation to precession cycles.

Supplementary Figure S1. Comparison of upwelling proxies at LV76-16-1 and IODP Site U1343 with boreal summer insolation. (a) Reconstructed upwelling index of core LV76-16-1 (this study; black line) and summer insolation at 65°N (Berger and Loutre, 1991; red line). **(b)** Upwelling index of IODP Site U1343 from the Bering Sea (Worne et al., 2019; blue line) and summer insolation at 65°N (Berger and Loutre, 1991; red line). The vertical green dashed lines represent 70 ka, 95 ka, 190 ka and 210 ka, four dates which were explicitly mentioned by Reviewer #2.

Reviewer #3

Comment:

This is a comment to the manuscript by Yao et al. and specifically the reviewer's comments, which show some opposite opinions after two rounds of review.

Reviewer 1 provides an extensive and constructive review, which is was answered in detail by Yao et al. Reviewer 2 has a problem with the age model presented by Yao et al. and argues that the paper by Galbraith et al. shows opposite results because of the age model problem. However, Yao et al. argue that the interpretation of the reviewer 2 was not correct and provide an explanation that appears valid.

What additionally argues for me that the age model, and therewith the story, is valid, is that their age model and trends in productivity on themselves are not something new for the north Pacific. These productivity peaks have been identified at many locations following a similar mechanism, including reference to the Galbraith et al. paper. This is also where their age model is based on, i.e. comparing these peaks to the neighboring ODP Site 882 and align them to construct the age model.

Obviously, and also determined by setting, the age model would not be valid for high-resolution conclusions on leads and lags during the termination. But Yao et al. are also staying away from this. One, as a minor side comment, thing that I noticed in the age model plot was that the variations in sedimentation rate are not very large due to the limited number of tie-points. Although difficult to constrain, these peaks in productivity probably increased sedimentation rates significantly during the time intervals that these conditions prevailed. But this would not change the correlation of the peaks with those at other locations in the north Pacific as these would have experienced the same changes.

As an important part of the paper, it is concluded that the precession has a significant impact on the cyclicity of the productivity and thus on CO₂ release. What would be interesting to include in figure showing atmospheric CO₂ is if this actually also contains the precession component. If the reservoir in the north Pacific is so large, this should be present in the overall signal as well.

Reply:

Thank you very much for your thorough and objective evaluation of our manuscript. We deeply appreciate the constructive comments and the time you have dedicated to our work. We agree with your perspective on the age model and are grateful for your support of our explanations against the concerns raised by Reviewer 2. Your deep understanding of our

approach reinforces the validity of our findings and the robustness of the age model we employed, particularly in the context of the North Pacific's productivity peaks and their correlation with climate changes.

Furthermore, in response to your insightful suggestion about examining the presence of a precessional cycle in atmospheric CO₂, we have added a figure (Extended Data Figure 4d shown below) in the revised manuscript. This figure presents a wavelet analysis of CO₂ variations, and our results indeed show that CO₂ changes contain a significant precessional component. This finding further strengthens our argument regarding the significant impact of precession on productivity cycles and consequent CO₂ release, underscoring the critical role of North Pacific in regulating carbon cycles.

We believe that this additional analysis not only addresses your valuable feedback but also enriches the scientific discussion around our findings, providing a clearer picture of the intricate relationship between orbital forcing, productivity, and carbon cycles in the North Pacific.

Extended Data Fig. 4. Variations and wavelet analysis of Ba/Ti, upwelling index and $p\text{CO}_2$.

(a) Comparison between Ba/Ti (dark line), 20-kyr filtering (blue line), Precession (red line) and Eccentricity (green line)⁶⁰. (b) Wavelet analysis of Ba/Ti from core LV76-16-1. (c) Wavelet analysis of upwelling index from core LV76-16-1. (d) Wavelet analysis of $p\text{CO}_2$ ^{1,66}.

References

- Berger, A. & Loutre, M-F. Insolation values for the climate of the last 10 million of years. *Quat. Sci. Rev.* **10 (4)**, 297–317 (1991).
- Galbraith, E. D. et al. Consistent relationship between global climate and surface nitrate utilization in the western subarctic Pacific throughout the last 500 ka. *Paleoceanography* **23**, 1–11 (2008).
- Jaccard, S. L. et al. Glacial/interglacial changes in subantarctic North Pacific stratification. *Science* **308**, 1003–1005 (2005).
- Worne, S., Kender, S., Swann, G. E. A., Leng, M. J. & Ravelo, A. C. Coupled climate and subarctic Pacific nutrient upwelling over the last 850,000 years. *Earth Planet. Sci. Lett.* **522**, 87–97 (2019).